# Minimal Hepatic Encephalopathy Affects Daily Life of Cirrhotic Patients: A Viewpoint on Clinical Consequences and Therapeutic Opportunities

**DOI:** 10.3390/jcm11237246

**Published:** 2022-12-06

**Authors:** Jessica Faccioli, Silvia Nardelli, Stefania Gioia, Oliviero Riggio, Lorenzo Ridola

**Affiliations:** Department of Translational and Precision Medicine, Sapienza University of Rome, 00185 Rome, Italy

**Keywords:** minimal hepatic encephalopathy, cirrhosis, quality of life, sleep disorders, therapy, non-absorbable disaccharides, rifaximin

## Abstract

Minimal hepatic encephalopathy (MHE) is a frequent complication of hepatic encephalopathy (HE) and can affect up to 80% of patients with liver cirrhosis. It is characterized by the lack of obvious clinical signs and the presence of alterations detectable using psychometric or electrophysiological testing focused on attention, working memory, psychomotor speed and visuospatial ability. Ideally, each patient should be tested for this condition because, despite the absence of symptoms, it has severe repercussions on daily life activities. It may be responsible for an inability to drive, sleep disturbances, risk of falls and inability to work. Some studies have highlighted its prognostically unfavorable role on mortality and risk of “overt” HE (OHE). Finally, MHE severely affects the lives of patients and caregivers, altering their quality of life and their socioeconomic status. Several treatments have been proposed for MHE treatment, including non-absorbable disaccharides, poorly absorbable antibiotics, such as rifaximin, probiotics and branched-chain amino acids, with promising results. For this reason, early diagnosis and intervention with appropriate measures is essential, with the aim of improving both performance on psychometric tests, as well as clinical aspects related to this condition.

## 1. Introduction

Hepatic encephalopathy (HE) is a neurocognitive disorder associated with both acute and chronic liver damage in which brain function is impaired. 

It is a frequent complication and one of the most debilitating clinical manifestations of liver disease, associated with decreased survival and high risk of recurrence. 

It is defined as a brain dysfunction related to liver failure and/or the presence of portosystemic shunts, and it is characterized by a wide spectrum of neurological and psychiatric changes, ranging from subclinical alteration to more severe forms that may begin with hepatic coma. 

The prevalence of HE is generally 10–14% at the time of diagnosis of cirrhosis, 16–21% in those with decompensated cirrhosis, and 10–50% in patients with transjugular intrahepatic portosystemic shunt (TIPS). The risk of a first episode of HE in the first 5 years after diagnosis of cirrhosis is 5–25%, while in those with previous episodes of overt HE (OHE), the risk of one year recurrence is 40% [1]. 

Its occurrence causes a worsening of prognosis; in fact, patients with cirrhosis and HE have double the one year mortality risk of patients with cirrhosis without HE [2].

Patients with a higher risk of HE are those with decompensated cirrhosis, spontaneous or iatrogenic shunts, previous OHE, minimal HE (MHE) and muscle changes, such as sarcopenia [3].

Patients with OHE exhibit temporo-spatial disorientation and inappropriate behaviors; appear agitated or, conversely, drowsy; and may progress to deep coma.

Clinical severity is defined according to the West Haven scale. In grade II, patients are disoriented in time; in grade III they are disoriented in space and time with Glasgow coma scale (GCS) > 8; in grade IV, GCS is < 8 and patients do not respond to painful stimuli [2].

Diagnosis is based on excluding other causes of altered mental status because symptoms may resemble those of other neuropsychiatric conditions. 

So, it is critical to confirm the presence of cirrhosis and to assess its severity using appropriate scores (i.e., Child–Pugh, MELD score). Plasma ammonium dosage may be useful to rule out the diagnosis. 

The second step is to exclude other causes of neuro-psychiatric symptoms, such as acute alcohol withdrawal, hydro-electrolyte imbalances, drug abuse and psychiatric disorders with appropriate tests [1]. 

Brain imaging is necessary when clinical presentation is unusual, in case of diagnostic doubt or in case of treatment failure; in fact, if symptoms respond to therapies that reduce ammonium levels, it is likely to be HE [2,4]. 

The treatment of HE is based on adopting general measures for patients with altered consciousness, identifying and treating alternative or coexisting causes of altered mental status and any precipitating factors, and initiating empirical treatments to reduce ammonium levels. The main drugs used for this purpose are non-absorbable disaccharides, such as lactulose, and non-absorbable antibiotics, such as rifaximin. Other possible medications or strategies include polyethylene glycol, branched chain amino acids, probiotics and fecal microbiota transplantation [2]. 

## 2. Minimal Hepatic Encephalopathy

MHE is the mildest form of HE and is characterized by cognitive and psychomotor deficits without clinically recognizable symptoms of HE [5]. This condition was first described in 1970 by Zeegen et al. who noted that 38% of patients subjected to portal decompression surgery had abnormal “trail making test” scores [6]. Eight years later, the term subclinical HE was introduced and several definitions have followed since then. The latest classification combines MHE with grade I HE (according to West Haven criteria) under the term “covert” HE [7]. 

The incidence of MHE is 20–80%, and several studies have suggested that most patients with cirrhosis develop HE at some point during the course of the disease. This wide variability is related to different diagnostic criteria and different tests used for its recognition. 

Risk factors for MHE include advanced age, alcoholic etiology, previous HE, a more severe disease according to Child–Pugh and the presence of portosystemic shunts [8,9]. In addition, some etiologies increase the risk of cognitive decline, such as hepatitis C virus (HCV) infection [10] and diabetes mellitus, likely due to the negative effect on gastrointestinal motility [11].

Theories about pathogenesis of HE are numerous, but their validity has not yet been fully established. However, there is a general consensus about the impairment of blood neurotoxins detoxification due to liver failure and/or the presence of portosystemic shunts, resulting in changes in brain neurotransmission. Among the neurotoxins implicated in the pathogenesis of HE, the most important is the ammonia NH3/ammonium NH4+ system. 

Ammonium derives from the intestinal deamination of glutamine and bacteria metabolism of nitrogenous substances; under normal conditions, it is metabolized into urea through the liver urea cycle. In patients with cirrhosis, the enzyme activities are reduced and blood ammonium levels increase. 

Skeletal muscles may play a compensatory role in these patients because muscle glutamine synthase can metabolize ammonia into glutamine. Consequently, muscle depletion may favor ammonia accumulation and HE development [1]. 

Ammonia induces modifications of astrocytes structure, called type II astrocytosis, in which astrocytes assume a swollen shape with a nucleus broad and pale, prominent nucleolus and chromatin margination. 

The consequences of hyperammonemia can be associated with those of other neurotoxic substances of intestinal origin, such as mercaptans, derived from intestinal methionine metabolism, which are able to increase brain ammonium level; indoles, derived from tryptophan metabolism; short-chain fatty acids from anaerobic intestinal fermentation; and substances similar to benzodiazepines and manganese [12,13].

The effects of MHE depend on specific deficits, such as those in attention, vigilance and orientation, which can cause impairments in learning and working memory. In contrast, these deficits do not involve the verbal and communicative aspects, so patients are unaware of their symptoms.

Despite the absence of clinical symptoms, MHE is considered clinically relevant for at least three reasons. Firstly, it impairs everyday activities and has a negative effect on health-related quality of life (HRQOL) not only for patients but also for caregivers, resulting in increased use of health care resources [14]. Secondly, it predicts the risk of developing OHE, and finally it is associated with a worse prognosis [8]. 

Despite this, a 2007 survey by the American Association for the Study of Liver Diseases (AASLD) showed that most physicians believed that MHE was a significant problem while also remaining under-investigated. In fact, only 50% of physicians evaluated cirrhotic patients for MHE and as many as 38% had never performed a psychometric evaluation of these patients [15].

Therefore, taking these factors together, early diagnosis and eventual treatment are crucial. 

## 3. Diagnosis of Minimal Hepatic Encephalopathy

Each patient should be tested for MHE at the time of diagnosis and later during follow-up because it constitutes a serious health problem and, despite its “minimal” expression, is associated with poor prognosis and quality of life. 

The diagnosis of MHE can be made with psychometric (computerized and non-computerized) and electrophysiological tests.

Electrophysiological tests are based on standard methodology, require sophisticated equipment and have lower sensitivity than psychometric tests. Computerized tests, on the other hand, are generally based on repeating a large number of tests and give more accurate results than “paper-pencil” tests.

Tests used for diagnosing MHE are: -Animal naming test (ANT): This is useful as a screening test. Patients have to list as many animals as possible in one minute and a number of animals <15 is indicative of MHE; it is conditioned by education level (<8 years) and age (>80 years);-Psychometric hepatic encephalopathy score (PHES): This is considered the gold standard for MHE diagnosis (Figure 1). It includes a battery of “paper-pencil” tests for the assessment of psychomotor speed and skill, set shifting, attention, visuospatial orientation, concentration and memory; it lasts about 15 min and a score <−4 is indicative of MHE;-Critical flicker frequency (CFF): he has to press a button as soon as the impression of fused light switches to oscillating light; it takes about 10 min and it is useful for the evaluation of visual apparatus and cerebral cortex;-Continuous reaction time test (CRT): The patient has to press a button in response to one-hundred 500 Hz tones presented at 90 dB in random intervals. CRT-index expresses the variability of response times and state of alertness. -Inhibitory control test (ICT): Several letters are presented at 500 msec intervals, with X and Y interspersed within these letters; patients have to respond only when X and Y are alterning (targets) and not when X and Y are non-altering (lures). This test assesses attention and inhibition (Figure 2);-Stroop test: In the OFF state, the patient sees a neutral stimulus and has to respond as soon as possible by touching the matching colour of the stimulus to the colour displayed at the bottom of the screen. In the ON phase, the patient sees discordant stimuli and has to touch the colour of the word presented, which is the name of the colour in discordant colouring. This test assesses attention; -EEG (electroencephalogram): This is useful for studying the cortical activity. In patients with OHE, cerebral activity is slower and three-phase waves are observed [16]. In patients with MHE, the quantative EEG (q-EEG) shows an increase in theta band and a decrease in the MDF (mean dominant frequency) in the posterior derivations, and changes in MDF during sleep represent early markers of brain disfunction. The q-EEG analysis shows alterations in slow oscillatory activity, with an increase in the frequency of dominant delta-rhythm. Evoked potentials: P300 wave (elicited by decision making) has lower amplitude and frequency in MHE [2]; -Magnetic resonance imaging (MRI), MR spectroscopy (MRS) and artificial intelligence: The use of MRI with voxel-based morphometry analysis in patients with liver cirrhosis for the assessment of brain density reveals a reduction in both white and gray matter, mainly in patients with alcoholic etiology, previous OHE and MHE. Such alterations seem to persist even after liver transplantation [17]. -MRS allows the measurement of different metabolites in the brain. In patients with HE, this analysis showed a reduced concentration of myoinositol, increased level of glutamine and decreased choline peak intensities. The elevation of cerebral glutamine concentration is probably do to hyperammonemia, while the lower concentration of myoinositol suggests an important osmoregulatory activity within the astrocyte. These alterations seem to correlate with the grade of HE, as well as with the performance in neuropsychological tests [18,19]. Recently, artificial intelligence has been introduced into many areas of medicine and clinical practice. Several studies have been performed regarding its role in hepatology and MHE diagnosis, mostly using brain magnetic resonance imaging but only in research settings. Machine learning-based approaches role is to examine functional magnetic resonance imaging (fMRI) data in a multivariate manner and extract features predictive of group membership. In particular, using information regarding microstructural integrity and water movement through cell membranes of white matter and the total grey matter volume; machine learning can discriminate cirrhotic patients with and without MHE. However, the costs associated with these technologies are high and currently not sustainable in clinical practice [20]. 

Unfortunately, there is no single optimal diagnostic method because each method explores different brain functions and none of them cover all aspects of HE. So, as MHE may affect different cognitive domains in different patients, a useful approach might be to use a combination of tests covering the major aspects involved. A possible strategy for MHE diagnosis is to screen cirrhotic patients with rapid and highly sensitive computerized psychometric tests, and then use PHES for further validation [2]. 

Appendix A describes the main tests used for MHE diagnosis and their characteristics. 

## 4. Minimal Hepatic Encephalopathy and Quality of Life

Quality of life is a multidimensional concept, affecting aspects of human well-being and encompassing physical and cognitive abilities, functional behavior, emotional state and psychosocial regulation [9]. 

In patients with liver cirrhosis, it is related to a potentially treatable factor; so, it is a key component in evaluating the effectiveness of various therapeutic interventions.

Quality of life assessment can be performed using questionnaires:-Sickness impact profile (SIP): this questionnaire is based on 136 items grouped into 12 scales (sleep and rest, eating, work, housekeeping, recreation and hobbies, walking, mobility, body care and movement, social interaction, vigilance, emotional behavior and communication) and provides a total score ranging from 0, corresponding to the best emotional state, to 100, corresponding to the worst one [21];-Chronic liver disease questionnaire (CLDQ): This contains twenty-nine items grouped into six domains that include abdominal symptoms (three items), fatigue (five items), systemic symptoms (five items), activity (three items), emotional function (eight items) and worry (five items). For each question, patients use a 7-point scale; higher scores indicate better quality of life [22];-Short Form-36 (SF-36): This is a paper–pencil test adjusted for age and educational level. This test measures eight domains; four related to “physical health” (physical functioning, physical limitation, physical pain and general health) and four related to “mental health” (vitality, mental health and social functioning). Each domain has a score from 0 to 100 and higher scores indicate better quality of life. However, it has only been validated in the Italian population [23,24];-Nottingham Health Profile (NHP).

Several studies have shown that liver disease affect health-related quality of life (HRQOL).

Activities most affected are those that require attention, information processing and psychomotor skills, such as driving a car or planning a trip. In contrast, everyday activities, such as personal hygiene, dressing or shopping, are preserved [8]. 

Several mechanisms can be responsible for the impairment of HRQOL in cirrhotic patients. Patients with advanced disease have to limit their daily activities due to impaired physical performance. Similarly, complications of cirrhosis, such as ascites and bleeding, impair HRQOL and social interactions.

Marchesini et al., using SF-36 and NHP, showed that cirrhosis etiology and disease duration had no effect on HRQOL, while symptoms, such as itching and muscle cramps, strongly affected it [25]. Hospitalizations, coexistence of sleep disturbances and disease severity also have a role in determining quality of life [26,27]. Finally, the study by Nardelli et al. showed that symptoms, such as depression, alexithymia and anxiety, were frequent among cirrhotic patients and were among the major determinants of HRQOL [28]. 

There is convincing evidence on the role of the cognitive decline on patients’ daily activities, well-being and HRQOL. 

Groenweg et al. showed that patients with MHE had significant reduction in all 12 SIP scales, psychosocial sub score, physical sub score and total score, compared with patients without MHE. On multivariate analysis, the presence of MHE was correlated with reduction in SIP scale, mostly in the area of vigilance, social interactions, recreation and work [29]. 

A study published by the same group, showed that the SIP statements predictive for the presence of MHE were: vigilance (forgetfulness, confusion, alertness), sleep and rest (sleeping, dozing during the day), fine motor activities and work. Regarding this last item, 50% of patients with MHE reported not having regular employment, compared with 15% of patients without MHE [29], similarly to Schomerus et al. who showed that nearly half of patients with MHE were unable to work [30].

Mina et al., regarding the role of MHE on patients’ HRQOL, explored for the first time the role of appetite in patients with MHE [31]. They found that patients with MHE had reduced appetite, and Child–Pugh score was a risk factor for loss of appetite and reduced HRQOL. So, this study showed that both MHE and loss of appetite have a negative effect on HRQOL in patients with decompensated liver cirrhosis [31].

These findings increase interest in demonstrating whether specific treatment of MHE could result in improvement of HRQOL [25,29,32]. Minimal hepatic encephalopathy and daily life functioning

### 4.1. Sleep Disorders

Sleep is a complex and highly regulated process, fundamental for human health and well-being. Increasingly, sleep–wake cycle disorders seem to be implicated in the pathogenesis of chronic liver disease. 

Sleep abnormalities have a prevalence of 26–70% among cirrhotic patients, with substantial repercussions on quality of life and physical health [33,34]. Moreover, sleep abnormalities can worsen patient’s liver function and cognitive status [35]. 

Patients with sleep disorders may complain of several types of symptoms: -Insomnia, difficulty getting to sleep or difficulty staying asleep; -Hypersomnia or excessive sleepiness; -Unusual events associated with sleep, such as apnea or abnormal movements [36].

Sleep assessment can be performed subjectively or objectively. Subjective assessment is based on the use of diaries or questionnaires:-Pittsburgh Sleep Quality Index (PSQI): This is the gold standard for the assessment of the previous month’s sleep, and it lasts about 15 min. It includes 19 items grouped into seven groups (sleep quality, sleep latency, sleep duration, sleep efficiency, sleep disturbance, sleep medication and diurnal dysfunction). For each group is assigned a score from 0 to 3 and their sum results in a total score between 0 and 21. A score > 5 indicates poor sleep quality; -Sleep Timing and Sleep Quality Screening Questionnaire (STSQS): This is a simplified and faster form of the previous one, taking only 2 min. It provides information on sleep quality and time, such as the time you go to bed, latency, night awakenings and wake up in the morning;-The Epworth Sleepiness Scale (ESS): this assesses daytime sleepiness in eight different situations (while reading, in front of TV, sitting in a public place, passenger in a car, while stopped in traffic, afternoon rest, while talking to someone and sitting after a meal).

Objective assessment is performed with special tools:-Polysomnography (PSG): this represents the gold standard because it assesses brain electrogenesis, eye and skeletal muscle movements, blood oxygen level and respiratory rhythms during sleep. However, it is expensive and time consuming, so it is generally used for research purposes.-Actigraphy: this is a semi-quantitative technique that uses an actigraph, which is a three-dimensional sensor placed on a wrist that records patients’ movements. Actigraphy assesses periods of quiet and movement over one or more days, considering that if a patient is awake there are movements, while if he is asleep, they are absent. Information obtained are related to total sleep duration, sleep latency and number of nocturnal awakenings [37]. 

Mechanisms responsible for sleep alterations in patients with liver cirrhosis are numerous. 

The clearance of melatonin is reduced because of its hepatic metabolism, and this results in excessive daytime plasma levels. In addition, the alteration of the circadian rhythm of melatonin would seem to be involved with a delay in its nocturnal plasma peak, probably due to reduced sensitivity to light [36]. Additional mechanisms are neuromuscular and thermoregulatory alterations. 

Although there are numerous studies on sleep disorders in patients with liver cirrhosis, those related to HE are scarce.

The earliest evidences in favor of a link between HE and sleep disorders came from Sherlock et al. who noted that sleep–wake cycle inversion, restless nights and excessive daytime sleepiness represented early symptoms of OHE [38]. 

Moreover, the evidence that sleep disturbances may occur after TIPS (transjugular intrahepatic portosystemic shunt) placement may suggest a common pathogenetic mechanism between HE and sleep alterations [39].

Confirming this hypothesis, sleep disturbances seem to develop in line with changes in ammonium plasma levels, which have a key role in the pathogenesis of HE [40]. 

Finally, some electroencephalographic changes observed during episodes of HE, such as the anteriorization of the alpha rhythm, are similar to those observed during the transition from wakefulness to sleep [36]. 

Insomnia is the most frequent sleep disorder, even during disease compensation; however, some studies did not show significant differences between cirrhotic patients with and without HE. 

In the study by Montagnese et al., cirrhotic patients presented less restorative sleep than healthy controls, without significant differences between sleep indices and the presence/grade of HE, although demonstrating a negative impact on HRQOL [34].

On the contrary, other studies demonstrated a positive association between sleep disturbances and HE. The absence of excessive daytime sleepiness seem to have a negative predictive value on the risk of hospitalization for HE [41], while its occurrence could be responsible for lack of restorative sleep [36].

In this regard, in the study by Samanta et al. in which 100 cirrhotic patients were enrolled and divided between those with MHE (*n* = 46) and those without (*n* = 54), 60% were “poor sleepers” according to the PSQI, while 38% demonstrated excessive daytime sleepiness, as measured by the ESS. In terms of multivariate analysis, MHE resulted in a strong correlation with nocturnal sleep disturbances and excessive daytime sleepiness, and both of these conditions resulted in associations with impaired HRQOL. In fact, MHE was found in 87% of “poor sleepers” and in 6% of “good sleepers” [33]. 

Finally, in a study conducted in India, both excessive daytime sleepiness and poor nocturnal sleep quality were found more frequently in cirrhotic patients with MHE than those without MHE [42]. 

MHE seems to cause alterations in sleep architecture. Sleep architecture refers to the repetition of REM (rapid eye movements) and NRME (non-REM) sleep cycles; the latter is further divided into four stages. In patients with normal sleep architecture, the length of stage 1, stage 2, slow waves sleep (SWS) and REM is 5–10%, 50%, 20%, and 20–25% of sleep duration, respectively. The NREM sleep component increases attention, logical thinking ability, language and foresight, and it can increase adaptive capacity and flexibility in response to environmental changes. The REM sleep component is strongly correlated to memory because it can promote plasticity.

Bajaj et al. firstly demonstrated altered sleep architecture among cirrhotic patients with MHE, compared to healthy controls. Specifically, they observed the absence of SWS in 80% and the increase in REM sleep time (19% vs. 7%, *p* = 0.02) in patients with MHE [43]. 

Similarly, the study by Liu et al. proved a longer duration of stage 1 and 2 of NREM sleep, longer sleep latency, shorter REM sleep latency and higher frequency of nocturnal micro-waking. Finally, the rate of sleep maintenance and sleep quality were lower in patients with MHE than in the healthy controls, indicating a condition of dyssomnia and lower sleep quality [44].

Since sleep architecture is regulated by several cerebral centers, such as the hypothalamus, thalamus and pre-optic region, changes to this architecture could reflect the presence of MHE-induced neuronal damage [44]. 

Thus, in light of these data, the routine assessment of sleep quality and quantity and excessive daytime sleepiness is essential and this last condition should encourage research for HE [36]. 

### 4.2. Falls

Patients with liver cirrhosis have increased risk of falls [45]. Possible causes include endocrine alterations, such as hypogonadism, sleep problems, use of medications such as antidepressants, malnutrition and altered body composition, such as sarcopenia and myosteatosis, and cognitive decline. 

Such falls often cause fractures, hospitalizations, increased health care costs and worsening HRQOL. The consequences of falls can be relevant in cirrhotic patients because of coagulopathy and operative risk. 

Fall risk assessment can be performed by timed up and go test (TUG). During this test, the time taken to start from a sitting position, walk three meters, turn around and return to the original sitting position is measured. A value greater than 14 s was found to be indicative of a high risk of falls. 

In the prospective study by Soriano et al., patients with cognitive dysfunction had a higher incidence of falls (40.4% vs. 6.2%, *p* < 0.001) and a higher severity of falls with more frequent hospitalization (9.5% vs. 0%, *p* = 0.01). In the same study, cognitive dysfunction identified by PHES was predictive of falls at multivariate analysis (OR = 10.2, *p* < 0.001) [46]. 

Similarly, in the retrospective study by Roman et al., falls were more frequent in patients with MHE than controls and cirrhotic patients without MHE (40% vs. 12.9%, *p* > 0.001), with greater demand for medical care (8.8% vs. 0%, *p* = 0.004). Confirming this evidence, MHE (OR 2.91, *p* = 0.02), previous HE and antidepressant therapy resulted in associations with falls at multivariate analysis [47].

In previous studies, the risk of falls has always been related to cognitive decline and MHE. 

As is known, there is a close association between altered body composition and HE. 

Up to 40–70% of cirrhotic patients present sarcopenia, which is a wasting of skeletal muscle mass [48]; they may also develop an increase in intramuscular and intermuscular fat, a condition known as myosteatosis. 

These two conditions can coexist, albeit infrequently, and both are associated with increased risk of OHE and reduced survival [49]. 

The link between muscle changes and cognitive decline is the metabolism and trafficking of ammonium. This product has a key role in the pathogenesis of cognitive decline in cirrhotic patients due to the inability to remove ammonia through urea synthesis. In these cases, skeletal muscles can compensate for this defect by serving as alternative sites for urea clearance through glutamine synthesis. Thus, when a qualitative or quantitative change in skeletal muscle is present, this compensatory mechanism fails and HE is promoted [50].

So, it is possible that both cognitive and muscular alterations may increase the risk of falls. 

The study by Nardelli et al. aimed to understand whether these two conditions, MHE and muscular alterations, had a similar impact on a cirrhotic patient’s risk of falls. In this study, the prevalence of MHE was significantly higher in patients with previous falls and high risk for falls, according to a TUG test value > 14 s. The prevalence of myosteatosis was also higher in these patients. In fact, PHES, beta blockers use and muscle attenuation resulted in a significant correlation with risk of falls. The incidence of falls during follow-up was significantly higher in patients with myosteatosis, but not in those with sarcopenia, and was higher, but did not reach statistical significance, in those with MHE. Thus, muscle alterations seem to play a greater role on risk of falls than altered cognitive status [51]. 

MHE may increase the risk of falls because it can induce movement disorders, which are still poorly studied in cirrhotic patients. 

The study by Urios et al. showed that patients with MHE have increased risk of falls due to impaired postural control, reduced stability, increased reaction time and delayed onset of movements [52]. Data suggest that this last characteristic may be the major cause of bradykinesia in MHE [53]. 

In a recently published cross-sectional study comparing cirrhotic patients with and without MHE by San Martín-Valenzuela et al., a biomechanical assessment of gait, balance, hand strength and speed in manual actions, was performed. Cirrhotic patients with MHE were slower, with poorer balance, a longer support phase and less pushing force from the ground before the swing phase. They also had greater variability in motor reaction time and lower efficiency in manual activities [54].

Thus, this evidence seems to support the direct relationship between MHE and falls, involving muscle changes and movement disorders. So, health care providers, including nurses and physical therapists, need to be aware of the increased risk of fractures in these patients, and both therapeutic and preventive strategies need to be implemented, especially in those awaiting liver transplantation.

### 4.3. Ability to Work and Wages

Because MHE impairs many cognitive functions, it may have a negative effect on patient’s ability to work, depending on the type of work performed.

So-called “white collar” workers are those workers with intellectual functions, not directly involved in productive activity. In contrast, “blue collar” workers correspond to those workers who perform more manual activities.

The cognitive impairment induced by MHE, causes an imbalance in the work ability of cirrhotic patients. In fact, the “blue collars”, who are mainly employed in activities that require vigilance and manual dexterity, have a disadvantage compared to those who perform predominantly intellectual activity. 

Shomerus et al. analyzed 110 cirrhotic outpatients who were previously judged fit for work by an expert. Despite this, 44% of patients were judged unfit for work, with a clear difference between “blue collar” and “white collar” (60% vs. 20%). No significant differences were found between employed and unemployed in terms of severity of illness, while a significant difference was found about psychometric function [30]. 

Reduced work capacity has a significant socio-economic impact on patients and caregivers. Indeed, reduced work performance and lost wages increase the indirect costs associated with this condition [55]. Finally, reduced work efficiency represents a potential danger for patients and colleagues.

### 4.4. Driving Skills

Driving is a complex and potentially dangerous function, involving the integration of visuomotor coordination, orientation and selective attention. Visuomotor coordination is given by all the information and stimuli coming from traffic, road signs and traffic lights. Orientation is an executive function involving planning, decision making, calculation of potential errors and inhibitory response. 

Patients with MHE may be at risk of road accidents because they have impaired attention and delayed reaction times, conditions that may predispose them to the difficulty of controlling a vehicle. 

Several studies have been carried out with the aim of studying this relationship.

Assessment of driving ability can be performed using several methods: -Neuropsychological assessment of cognitive domains involved in driving activity;-Virtual simulators: i.e., SIMUVEG driving simulator, STISIM simulator. During the simulation, which lasts > 10 min, several variables related to roads, times, distances, actions and decisions are considered to evaluate the driver’s driving ability. Two essential aspects of driving performance are longitudinal and lateral control of the vehicle. The former is related to average speed, while lateral control is related to angular speed, wheel movement, distance and time-to-line crossing [56,57];-Road tests: The assessment is performed by a professional driving instructor who is unaware of the subjects’ diagnosis and test results. The assessment is based on the evaluation of four driving categories: car handling, adaptation to traffic situations, caution and vehicle maneuverings. The driving instructor uses a point rating scale to judge driving competence for each category and gave a final score for the overall impression [58].

In the prospective study by Wein et al., the assessment of driving ability by road test showed that the overall driving score of cirrhotic patients with MHE was significantly lower than that of controls or patients without MHE. The most impaired categories were car handling, adaptation, maneuverings and prudence. In addition, the instructor had to intervene more frequently to avoid an accident [58]. 

It was not known whether this impairment of driving skills increased the risk of traffic violations or accidents.

The study by Bajaj et al. showed that cirrhotic patients have more traffic accidents and commit more traffic violations, or both, than controls of the same age and level of education within 1 and 5 years of questionnaire administration. On multivariate analysis, the presence of MHE was predictive for their occurrence [59].

The same author, in a subsequent prospective study, including only cirrhotic patients, confirmed the previous finding by showing that those with MHE had higher rates of traffic accidents in the previous and subsequent year (22% vs. 7%, *p* = 0.03). Both MHE identified by ICT and accidents/traffic tickets in the past year were predictive of future accidents (OR = 4.51 and OR = 2.96, respectively) [60]. 

Patients are often unaware of this, tending to overestimate their driving skills. 

The study by Kircheis et al. showed that among cirrhotic patients with MHE and Grade I HE, 48% and 39%, respectively, were able to drive according to the driving instructor’s judgment, compared with 79% and 89% of cirrhotic patients without HE and healthy controls. Therefore, having MHE correlated with an increased risk of impaired driving ability. In this study, the agreement between instructor judgment and psychometric test results was poor, equal to 70%, with more severe evaluations by the instructor. Another interesting result of this study was that when subjects were asked to judge their driving ability, 100%, 97% and 90% of those without HE, with MHE and with grade I HE, respectively, rated it as good, thus demonstrating a tendency to underestimate the problem [61]. 

The study by Bajaj et al. also confirmed this finding, showing that patients with MHE judge themselves as capable as cirrhotic patients without MHE and healthy controls, despite having significantly worse driving performance [57]. 

In addition, recent evidence has proved that cirrhotic patients have worse outcomes after traffic accidents, with higher in-hospital mortality and longer hospitalization times, as well as higher healthcare costs [62]. 

Therefore, the determination of driving impairment requires ongoing investigation and definition as traffic accidents are important causes of morbidity and mortality. Because these patients are asymptomatic and often even unaware, it is critical to investigate the history of traffic accidents and traffic violations, collaborate with family members and educate patients as part of the medical interview to increase the likelihood of implementing targeted therapy.

## 5. Development of “Overt” Hepatic Encephalopathy and Prognosis

MHE is a recognized risk factor for progression to OHE and mortality [63]. More than 50% of patients with MHE develop episodes of overt hepatic encephalopathy (OHE) within three years [9].

The relationship between MHE diagnosis and development of OHE is extremely important because the first episode of HE is associated with a reduction in survival (23% at 3 years) [64]. 

In the prospective study by Hartmann et al., during a mean follow-up of 29 months, patients with MHE presented a 3.7-fold increased risk of developing OHE (*p* = 0.002) and more episodes of OHE during follow-up (56% vs. 8%, *p* < 0.001), compared to patients without cognitive decline. However, Child–Pugh score had a greater impact than MHE on OHE development during follow-up (RR 19.3%, *p* < 0.001). Conversely, no differences were found in mortality, which was mainly determined by Child–Pugh score (RR = 13.95%, *p* = 0.003) [65].

The predictive role of MHE on OHE development was confirmed by subsequent studies in which a higher prevalence of MHE was found among patients who developed OHE during follow-up, with higher ammonium level and greater disease severity identified by MELD [66,67]. 

In fact, the natural history of MHE is worse in patients with more impaired liver function. Among patients with MHE, the development of OHE is greater in those with advanced cirrhosis and Child–Pugh scores > 7 [9].

However, the clinical relevance of MHE is not only related to the risk of developing major complications but also to its impact on patient survival. There is a great variability in the literature on this topic in light of the different diagnostic tools used to define the presence of MHE. 

In this regard, Dhiman et al. defined cut-off values for both the Child–Pugh score and PHES score based on their relationship with survival. In fact, a PHES < −6 and a Child–Pugh score > 7 were associated with a significant increase in mortality [68].

Amodio et al. also showed that MHE, at multivariate analysis, had a prognostic value on survival during the first year of follow-up, together with disease severity identified by Child–Pugh score (HR = 2.4 for Scan test, *p* = 0.035) [69]. 

Thus, although MELD and Child–Pugh scores are currently used to assess the prognosis of the cirrhotic patients, it is clear that cognitive function also plays a determining role. Therefore, it is desirable for the future that new, more accurate staging systems will be introduced to consider this parameter as well.

## 6. Therapy of Minimal Hepatic Encephalopathy

Since the proposed pathogenetic mechanism of MHE is similar to that of OHE, the therapeutic strategies used for MHE are the same as those used for the treatment and prophylaxis of OHE. 

Strategies for the treatment of MHE should include both reduction in ammonium formation, such as through adequate energy and protein intake (30–45 kcal/kg/day and 1.2–1.5 g protein/kg/day, respectively), and enhancement of ammonium detoxification/elimination (fiber supplementation, nonabsorbable disaccharides and antibiotics). 

The main treatments for HE include nonabsorbable disaccharides, of which lactulose is the most widely used, and rifaximin.

Lactulose is fermented and metabolized into acetic acid and lactic acid; thus, the intestinal environment becomes acid and ammonia (NH3) and is converted into ammonium ions (NH4+), which cannot be absorbed through the intestinal barrier. This mechanism is also enhanced by its cathartic effect, which facilitates intestinal excretion of nitrogen. 

Several studies have evaluated the effect of lactulose on MHE, considering improved performance on psychometric tests as main endpoints.

The study by Watanabe et al. showed that lactulose treatment for 8 weeks resolved MHE in 50% of the 20 treated patients and persisted in 85% of the 13 untreated patients [70]. 

Similarly, Dhiman et al., after 12 weeks of lactulose therapy, demonstrated psychometric improvement in 80% of patients and in none of those who had not received such therapy [71].

Other studies came to the same conclusions regarding the efficacy of lactulose therapy [72]. Luo et al. found that lactulose therapy was superior to a placebo in all outcomes considered; in fact, it reduced the mean number of abnormal neuropsychological tests, time taken to complete the NCT-A (number connection test A), ammonium levels and risk of developing OHE, while improving quality of life [73]. 

Rifaximin is the second most widely used drug for HE. It is a non-synthetic and broad-spectrum intestinal antibiotic, which can modulate the intestinal production of ammonium and other toxins. 

The study by Zhang et al. showed that short-term therapy with rifaximin (one week) was able to resolve MHE based on psychometric test results, as well as to improve SIBO and plasma ammonium levels [74]. 

In a study of 20 cirrhotic patients with MHE by Bajaj et al., rifaximin significantly improved all but one of the psychometric tests used (NCT-A and B, DST, LTT, LTE, SDT) [75].

A comparison study between lactulose (30–120 mL/day) and rifaximin (400 mg three times a day) administered for 12 weeks published by Sidhu et al. demonstrated the effect of both drugs on psychometric test results, resulting in the disappearance of MHE in 69.1% of patients in the lactulose group and 73.7% of patients in the rifaximin group. However, the study did not demonstrate the non-inferiority of rifaximin to lactulose [76]. 

The combination of both drugs, after 8 weeks of therapy, significantly increased PHES score and reduced brain oedema in the study by Rai et al. [77]. 

Some studies have been performed on probiotics, which, by reducing the intestinal activity of bacterial ureases, decrease the absorption of ammonium and other toxins potentially involved in the pathogenesis of MHE.

Unfortunately, these studies have not demonstrated an advantage of probiotics alone or in combination with other drugs for the disappearance of MHE [78,79]. 

Although the relevance of these studies, one limitation is that they do not consider the clinical implications of the psychometric test and the effects of this therapy on major complications of MHE. 

Regarding the benefits of therapy on quality of life, Prasad et al. demonstrated that lactulose treatment in patients with MHE improved both cognitive performance and HRQOL and also that improvement of quality of life was related to a better performance on psychometric tests [8]. Similarly, Sidhu et al. demonstrated that rifaximin therapy improved the SIP score and resolved MHE in 75% of patients, with a positive correlation between improved performance on psychometric tests and HRQOL, compared with a placebo [80].

Quality of life also improved after administration of probiotics or LOLA (L-ornithine L-aspartate), in line with improvements in MHE [81]. Despite this, no significant differences in improvement in HRQOL, MHE, hospitalization, or risk of developing OHE were found when probiotic therapy was compared to lactulose [82]. 

Regarding the impaired driving skills of cirrhotic patients, Bajaj et al. showed that rifaximin therapy administered for 8 weeks versus placebo reduced errors in simulated driving (31% vs. 76%, *p* = 0.013), speeding (33% vs. 81%, *p* = 0.005) and illegal activities (19% vs. 62%, *p* = 0.01) while improving cognitive performance and quality of life in the psychosocial sphere [83].

In contrast to rifaximin, however, lactulose would be more cost-effective in reducing health care costs due to traffic accidents [84]. 

A number of studies have also been conducted regarding the role of therapy on risk of falls.

Roman et al. studied the therapeutic effect of probiotics; their use for 12 weeks in patients with MHE improved PHES and TUG test score and walking speed, as well as reducing inflammatory cytokines (PCR and TNF-alpha) and markers of intestinal permeability (FABP-6 and urinary claudin-3) [85].

Regarding sleep impairment, the administration of lactulose for 12 weeks determined the improvement of some parameters, such as total sleep time, sleep efficiency and latency, and waking time in patients with MHE [42]. 

However, there is no robust evidence that treatment of MHE reduces the risk of OHE. 

A study focusing on the use of probiotics vs. placebo for the treatment of patients with MHE showed that such therapy resulted in both the regression of MHE, with improvement on paper–pencil psychometric tests (71% vs. 0%, *p* = 0.003), and lower OHE during follow-up (0% vs. 25%), although this was not statistically significant [86]. Probiotics act by depriving substrates to potentially pathogenic bacteria and providing fermentation end products to beneficial ones.

Recently there has been growing interest in the role of albumin in patients with MHE. 

Fagan et al. recently published a randomized versus placebo clinical trial on the role of albumin on MHE and HRQOL in patients with prior HE and already on standard secondary prophylaxis. Albumin administration determined MHE reversal and improved quality of life, probably through improved endothelial dysfunction [87]. Finally, given the beneficial effects of therapies on MHE and as suggested by the most recent EASL guidelines, patients with MHE should be treated with nonabsorbable disaccharides and/or rifaximin, and, if effective, this may be an ex adiuvantibus criterion for diagnosis [7]. Appendix A summarizes the published study on the treatment of MHE [8,42,70,71,73,74,76,80,81,82,83,85,86,87,88,89,90,91,92,93,94,95]. 

## 7. Conclusions and Future Perspective 

MHE represents the earliest and mildest form of HE and it is often under-recognized and under-diagnosed. 

Although there is no gold standard for diagnosis, one or a combination of several psychometric and/or electrophysiological tests can be helpful for this purpose.

These tests, of which some are very simple for both examiner and patient, should be performed not only for research purposes but also at the time of diagnosis and then periodically during follow-up in all cirrhotic patients. Early diagnosis is crucial because an association with worse clinical outcomes has been demonstrated in patients with MHE. There is no gold standard for diagnosis of MHE; some tests are better than others, such as PHES, and so they serve as a reference for validating new tests. 

Moreover, a diagnostic tool able to identify the presence of MHE and assess its fluctuation over the time should be investigated. A diagnostic tool that, like a thermometer for a fever, can allow the clinician to follow the patient and monitor the effects of a proposed intervention(s), appears to be needed. Such a tool must be inexpensive, reproducible in results, easily accessible, non-invasive and safe.

However, despite the clinical implications of this condition, also due to the lack of a univocal diagnostic test, MHE is often missed. Therefore, it is crucial to develop effective diagnostic algorithms to progress screening and treatment. Moreover, multicenter studies are needed to explore the predictive value of these tests and the effects of associated co-morbidities. 

There are different options for treatment, such as non-absorbable disaccharides, rifaximin and probiotics. The management of MHE should also include a diet with adequate fiber, preferably plant-based protein, probiotics and physical exercise, if tolerated. The aim of these measures is to prevent malnutrition and sarcopenia and, therefore, to ameliorate ammonia catabolism, and thus improve the cognitive impairment that is typical of MHE. 

To date, the treatment of MHE is recommended by the recent EASL guidelines, and the two proven effective drugs are lactulose and rifaximin. There are numerous studies showing their effect not only on psychometric tests but also on the main clinical complications of MHE and quality of life. 

It is critical to conduct additional randomized clinical trials focused on robust endpoints to determine the best treatment strategies and their optimal duration, as well as to conduct a cost–benefit analysis of the various available therapies.

## Figures and Tables

**Figure 1 jcm-11-07246-f001:**
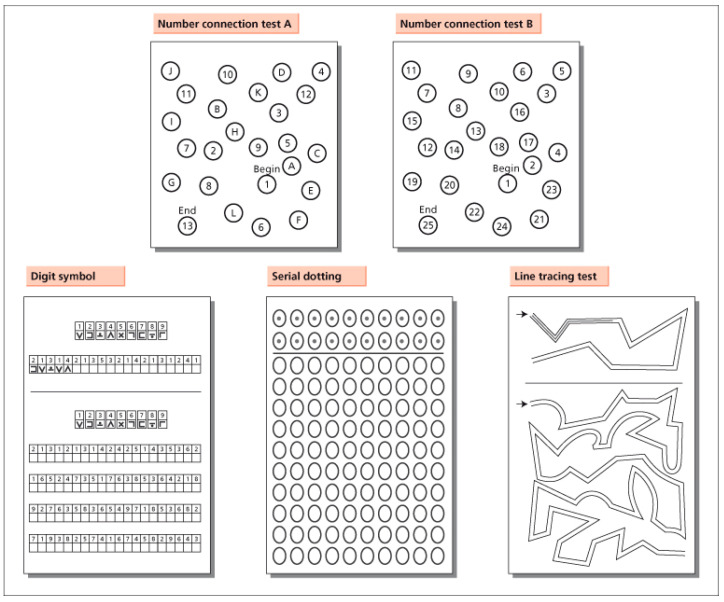
The psychometric hepatic encephalopathy score (PHES).

**Figure 2 jcm-11-07246-f002:**
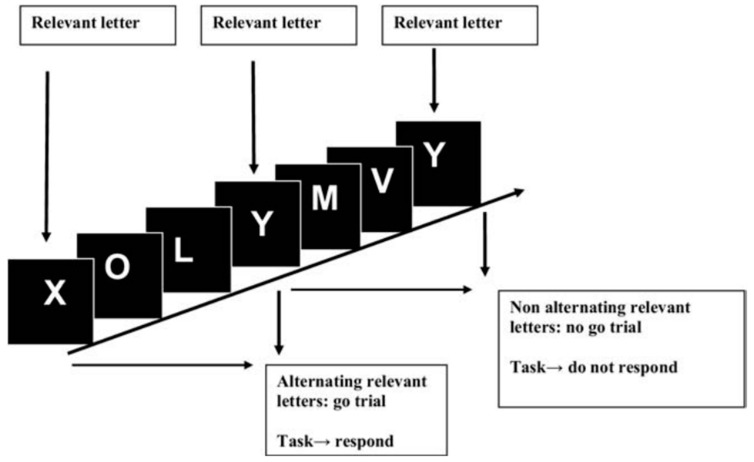
The inhibitory control test (ICT).

## Data Availability

Data sharing is not applicable to this article as no new data were created or analyzed in this study.

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
