# Peer review of "Minimal Hepatic Encephalopathy Affects Daily Life of Cirrhotic Patients: A Viewpoint on Clinical Consequences and Therapeutic Opportunities"

_jcm, 2022, doi:10.3390/jcm11237246_

Round 1

Reviewer 1 Report

The authors performed a review article on minimal hepatic encephalopathy, a clinically under-recognized condition, in cirrhotic patients. I have the following questions and comments.

1. Minimal hepatic encephalopathy has been well-described in the literature. It is often not considered to be clinically relevant, since diagnosed or treated MHE has not been shown to affect daily functioning, quality of life, driving and overall mortality. 

2. Is there a prisma systematic review flow diagram for all the included references?

3. After having done the review, how would the authors use the information to create an optimal, easy to use screening test for MHE? There are a few proposed screening mechanisms already published, ie EncephalApp Stroop Test, psychometric hepatic encephalopathy score, etc..

4. What would the authors proposed as a univocal diagnostic test for MHE? 

5. The article lacks applicability in clinical use and relevance regarding this topic, outside of what already been published. 

6. There are few misspellings throughout the article

Reviewer 2 Report

This review describes the clinical findings and treatment of mild hepatic encephalopathy in patients with cirrhosis, citing appropriate previous research articles.
The overall structure of the paper is very clear and is likely to attract the attention and citation of other researchers in the future.
The reviewers have several comments to make on this review article, and the authors would appreciate a brief response.

Major 1
JCM is a general medical journal, so please add an introduction that provides general knowledge of hepatic encephalopathy in a way that is easy for readers to understand. We would also like to see the epidemiological frequency and age at onset of hepatic encephalopathy, which are generally reported.

Major 2
The clinical diagnosis of MHE is described, but it is written only in text, so it is difficult to convey the image of MHE. For example, we would like to see an electroencephalogram (EEG) image of the standard triphasic waves seen in the EEG of OHE. Also, please provide pictures or images of other psychological tests that show the findings of encephalopathy cases, if any.

Major 3
Please add a prognosis for mild to severe hepatic encephalopathy.

Major 4
There is adequate description of medical treatments, but is there evidence for special therapies? For example, is plasma exchange for high ammonia effective? And how about gamma globulin therapy? Furthermore, is liver transplantation indicated as a surgical treatment?

Other

Minor1
Can you provide MRI images of the brain in cases of advanced encephalopathy?

Minor 2
What about differential diagnosis, such as dementia, normal pressure hydrocephalus, chronic subdural hematoma, congenital or acquired metabolic disorders?

Best regards, 

Dr. reviewer

Reviewer 3 Report

In this manuscript, Faccioli et al. described the clinical characteristics of minimal hepatic encephalopathy (MHE) in comparison with overt hepatic encephalopathy (OHE). However, this reviewer has the following concerns. 

Major comments: 

1. There are no figures and tables in this article. Some parts, for example, tests used for diagnosing MHE, had better be shown in the table.

2. Each paragraph is very short. It includes only a few sentences. This should be proofread. Regarding the sections, the new section ‘Symptom’ had better be added in front of the section, ‘Sleep disorders. Sections from ‘Sleep disorders’ to ‘Driving skills’ will be changed into subsections under the " Symptom " section.

3. Pathophysiology of MHE is lacking in this manuscript. It should be shown.

4. In the Therapy section, metabolic ammonia scavengers, such as sodium benzoate, glycerol phenylbutyrate, and ornithine phenylacetate should be described. 

Minor comments:

1.  On page 5, line244, “Montagnese et at” should be “Montagnese et al”.

2.  In the Introduction, lines 35-38, Regarding the description, “The incidence of MHE is 20-80%”, references should be added. 

Round 2

Reviewer 1 Report

The addition of images 1-4 is helpful to depict cognitive changes in patients with MHE.

The authors should consider concising the manuscript, as there is too much redundant contents. 

Author Response

Some sentences found to be redundant have been removed, as suggested. 

Reviewer 2 Report

I have reviewed the reviewer's substantially revised and revised resubmission of the manuscript. The content has been improved with the addition of pathology, EEG, and MRI as part of the encephalopathy review.

The reviewer still has one point to make.
Please use the comments below to revise again.

Major.
New Figure about MRI in Image3, but this image is not an MRI image. It is probably an MRI image or a cerebral blood flow image processed by image analysis software. We think it is necessary to add the official name of the image and the license of the analysis software. Authors should confirm this point and add it to the manuscript.

Minor
As a reviewer, I suggest that the authors revise Image 1, 2, 3, and 4 to Figure 1, 2, 3, and 4.

Best regards,

Dr. Reviewer

Author Response

As suggested, the requested changes have been made 

Reviewer 3 Report

All the comments have been addressed.

Author Response

Thank you for your consideration.